

# Are sites with multiple single nucleotide variants in cancer genomes a consequence of drivers, hypermutable sites or sequencing errors?

Thomas C.A. Smith[1], Antony M. Carr[2] and Adam C. Eyre-Walker[1]

[1] School of Life Sciences, University of Sussex, Brighton, East Sussex, United Kingdom
[2] Genome Damage and Stability Centre, University of Sussex, Brighton, East Sussex, United Kingdom

Corresponding authors
Thomas C.A. Smith,
tom81_asp@hotmail.com
Adam C. Eyre-Walker, a.c.eyre-walker@sussex.ac.uk

## ABSTRACT

Across independent cancer genomes it has been observed that some sites have been recurrently hit by single nucleotide variants (SNVs). Such recurrently hit sites might be either (i) drivers of cancer that are postively selected during oncogenesis, (ii) due to mutation rate variation, or (iii) due to sequencing and assembly errors. We have investigated the cause of recurrently hit sites in a dataset of >3 million SNVs from 507 complete cancer genome sequences. We find evidence that many sites have been hit significantly more often than one would expect by chance, even taking into account the effect of the adjacent nucleotides on the rate of mutation. We find that the density of these recurrently hit sites is higher in non-coding than coding DNA and hence conclude that most of them are unlikely to be drivers. We also find that most of them are found in parts of the genome that are not uniquely mappable and hence are likely to be due to mapping errors. In support of the error hypothesis, we find that recurrently hit sites are not randomly distributed across sequences from different laboratories. We fit a model to the data in which the rate of mutation is constant across sites but the rate of error varies. This model suggests that ∼4% of all SNVs are errors in this dataset, but that the rate of error varies by thousands-of-fold between sites.

## INTRODUCTION

There is currently huge interest in sequencing cancer genomes with a view to identifying the mutations in somatic tissues that lead to cancer, the so-called "driver" mutations. Driver mutations are expected to cluster in particular genes or genomic regions, or to recur at particular sites in the genome, because only a limited number of mutations can cause cancer. For example, the driver mutations in the TERT1 promoter were identified because it had independently occurred in multiple cancers (*Huang et al., 2013*). However, there are two other processes that can potentially lead to the repeated occurrence of an apparent somatic mutation at a site. First, it is known that the mutation rate varies across the genome at a number of different scales in both the germ-line and soma (*Hodgkinson & Eyre-Walker, 2011*; *Hodgkinson, Chen & Eyre-Walker, 2012*; *Michaelson et al., 2012*; *Francioli et al., 2015*).

Sites with recurrent SNVs could simply be a consequence of sites with high rates of mutations. Second, there is the potential for sequencing error. Although the average rate of sequencing error is thought to be quite low it is evident that some types of sites, such as those in runs of nucleotides, are difficult to sequence accurately. Furthermore, since the genome contains many similar sequences it can often be difficult to map sequencing reads successfully (*Treangen & Salzberg, 2013*).

In the germ-line the density of point mutations varies at a number of different scales (*Hodgkinson & Eyre-Walker, 2011*). At the mega-base scale the mutation varies by about 2-fold, and ∼50% of this variance can be explained by correlations with factors such as replication time, recombination rate and distance from telomeres (as reviewed in *Hodgkinson & Eyre-Walker (2011)*). However the greatest variance, reportedly up to ∼30-fold, has been found at the single nucleotide level (*Hodgkinson, Chen & Eyre-Walker, 2012*; *Kong et al., 2012*; *Michaelson et al., 2012*), whereby the nucleotide context, that is the identity of the bases immediately 5′ and 3′ of the mutated base, are highly influential on the rate of mutation (*Gojobori, Li & Graur, 1982*; *Bulmer, 1986*; *Cooper & Krawczak, 1990*; *Nachman & Crowell, 2000*; *Hwang & Green, 2004*). The most well known example is that of CpG hypermutation (*Bird, 1980*), which is thought to account for ∼20% of all mutations in the human genome (*Fryxell & Moon, 2005*). However there is also variation at the single nucleotide level that cannot be ascribed to the effects of neighbouring nucleotides; this has been termed cryptic variation in the mutation rate and is thought to account for at least as much variation in the mutation rate as does simple context (*Hodgkinson, Ladoukakis & Eyre-Walker, 2009*; *Eyre-Walker & Eyre-Walker, 2014*; *Johnson & Hellmann, 2011*; *Smith et al., 2016*).

The somatic mutation rate is estimated to be at least an order of magnitude greater than that of the germ line (*Lynch, 2010*). It has been shown to vary between cancers (*Lawrence et al., 2013*) and different cancer types are known to vary in their relative contributions of different mutations to their overall mutational compositions (*Alexandrov et al., 2013*). For a review see *Martinocorena & Campbell (2015)*. The aforementioned correlates of variation that are found in the germ line are also apparent in the soma (*Hodgkinson, Chen & Eyre-Walker, 2012*; *Schuster-Bockler & Lehner, 2012*; *Lawrence et al., 2013*; *Liu, De & Michor, 2013*), for example, replication time correlates strongly with single nucleotide variant (SNV) density at the 1 Mb base scale and can vary by up to 3-fold along the genome (*Hodgkinson & Eyre-Walker, 2011*; *Woo & Li, 2012*). However, as yet there has been no attempt to quantify the level of cryptic variation in the mutation rate at the single nucleotide level in the somatic genome. This is an important property to understand; for example, a site which experiences a recurrence of SNVs across many cancer genomes would be of interest as a potential driver of cancer (*Lawrence et al., 2013*), however, this site might simply be cryptically hypermutable (*Hodgkinson, Ladoukakis & Eyre-Walker, 2009*; *Eyre-Walker & Eyre-Walker, 2014*; *Smith et al., 2016*). Here we examine the distribution of recurrent SNVs taken from 507 whole genome sequences made publicly available by *Alexandrov et al. (2013)* to investigate the level of cryptic variation in the mutation rate for somatic tissues. We show that there is a large excess of sites that have been hit by recurrent SNVs. Since the density of these is greater in the non-coding than the coding fraction of the genome, we conclude that most of them are unlikely to be drivers. We therefore investigate
whether they are due to mutational heterogeneity or sequencing errors. In particular we investigate whether there might be cryptic variation in the mutation rate in cancer genomes. Unfortunately, the available evidence suggests that most sites with recurrent SNVs are likely to be due to sequencing error or errors in post-sequencing processing.

## METHODS

### Genome and data filtering

The human genome (hg19/GRCh37) was masked to remove simple sequence repeats (SSR) as defined by Tandem Repeat Finder (*Benson, 1999*). The remaining regions were separated into three genomic fractions, consisting of 1,346,629,686 bp of non-coding transposable element DNA (TE), defined as LINEs, SINEs, LTRs and DNA transposons as identified by repeat masker (RepeatMasker Open-3.0; http://www.repeatmasker.org/), 1,322,985,768 bp of non-coding non-transposable element DNA (NTE), and 119,806,141 bp of exonic non-transposable element DNA (EX) defined by Ensemble (*Flicek et al., 2012*). From the supplementary data of *Alexandrov et al. (2013)* we collated 3,382,737 single nucleotide variants (SNV), classified as "somatic-for-signature-analysis" (see *Alexandrov et al. (2013)* for SNV filtering methods). These can be downloaded from ftp://ftp.sanger.ac.uk/pub/cancer/AlexandrovEtAl/. These came from 507 whole genome sequenced cancers and represent 10 different cancer types and were reduced to 3,299,881 SNVs when excluding SNVs in SSRs; 1,666,759 in TE and 1,535,069 in NTE and 98,053 in EX.

### Testing for mutation rate heterogeneity

We were interested in whether some sites have more SNVs than expected by chance. Since the mutation rate is affected by the identity of the neighbouring nucleotides we need to control for those effects. To do this we separated each SNV into one of 64 categories based upon the triplet to which it was the central base. This was reduced to 32 triplets when accounting for base complementarity with the pyrimidine (C/T) taken as the central base. If the total number of triplets of type $i$ (e.g. C**T**C in the non-TE fraction) is $l_i$ and the number SNVs at that triplet is $m_i$ then the expected number of sites hit $x$ times can be calculated using a Poisson distribution:

$$P_i(x) = l_i \frac{e^{-\mu_i} \mu_i^x}{x!} \tag{1}$$

where $\mu_i = m_i/l_i$ is the mean number of SNVs per site, The expected number of sites with $x$ SNVs across all triplets was calculated by summing the values of $P_i(x)$. Whether the observed distribution deviated from the expected was tested using a chisquare test.

### Model fitting

As well as testing whether there was significant hetereogeneity we were also interested in quantifying the level of variation. We fit two basic models. In the first, we allowed the density of SNVs to follow a gamma distribution. Let the expected density of SNVs at a site be $\mu\alpha$ where $\mu$ is the mean density of SNVs for a particular triplet and $\alpha$ is the deviation

from this mean which is gamma distributed, parameterised such that the gamma has a mean of one. Under this model the expected number of sites with $x$ SNVs is

$$P(x) = l \int_0^\infty \frac{e^{-\mu\alpha}(\mu\alpha)^x}{x!} D(\alpha) d\alpha. \tag{2}$$

In a second model we imagine that the production of SNVs depends upon two processes, one of which is constant across sites, and one which varies across sites with the rate drawn from a gamma distribution. Let the proportion of SNVs due to the first process be $\varepsilon$. Under this model the expected number of sites with $x$ SNVs is

$$P(x) = l \int_0^\infty \frac{e^{-\mu(\varepsilon+(1-\varepsilon)\alpha)}(\mu(\varepsilon+(1-\varepsilon)\alpha))^x}{x!} D(\alpha) d\alpha. \tag{3}$$

Given the expected number of sites, the likelihood of observing $\widehat{P}(x)$ sites with $x$ SNVs is itself Poisson distributed

$$L(x) = \frac{e^{-P(x)}P(x)^{\widehat{P}(x)}}{\widehat{P}(x)!}. \tag{4}$$

These likelihoods can be multiplied across triplets to obtain the overall likelihood. We estimated the maximum likelihood values of the model parameters using the Maximize function of Mathematica which implements the Nelder–Mead algorithm (*Nelder et al., 1965*).

### Privacy analysis

To investigate whether the SNVs at some sites tended to be produced by a particular research group we took all sites with 3 or more SNVs from the same cancer type and then performed Fishers exact test on a $2 \times 30$ matrix using the the R stats package, version 3.2.4 (*R Core Team, 2016*).

### Mappability

Each nucleotide in genome was assigned a mappability score for uniqueness, as determined by the Mappability track (*Derrien et al., 2012*) downloaded from the UCSC table browser at http://genome.ucsc.edu/ (*Karolchik et al., 2004*). This feature assigns a value of 1 to unique $k$-mer sequences in the genome, 0.5 to those that occur twice, 0.33 to those that occur thrice etc. This is computed for every base in the human genome with the value being assigned to the first position of the $k$-mer. We used $k$-mers of 100 and 20 bases.

## RESULTS

### The distribution of recurrent SNVs

If there is no variation in the density of single nucleotide variants (SNVs) then we should find them to be distributed randomly across the genome. To investigate whether this was the case we calculated the expected number of sites with 1, 2, 3…etc SNVs, taking into account the fact that some triplets have higher mutation rates than others. We found that there are some sites that have 7 SNVs whereas we expect very few sites to have more than 3
**Table 1  Observed and expected values for the distribution of SNVs for sites hit from 0–7 times.** (A) shows data for the whole interrogable human genome, excluding simple sequence repeats. (B) shows data for all bases in the genome that are uniquely mappable at 100 base pairs. (C) the same as B but for 20 base pairs. $P < 0.001$ for observing >7 sites with 3 SNVs in (A), (B) and (C) if SNVs were randomly distributed throughout the genome.

| Site Type | 0 hits | 1 hit | 2 hits | 3 hits | 4hits | 5hits | 6hits | 7hits |
|---|---|---|---|---|---|---|---|---|
| **(A)–All Sites** | | | | | | | | |
| Non-Exon TE obs (TE) | 1344972042 | 1649680 | 7034 | 762 | 130 | 26 | 9 | 3 |
| Non-Exon TE exp (TE) | 1344964359 | 1663896 | 1430 | 1.14 | 9E−04 | 7E−07 | 5E−10 | 4E−13 |
| Non-Exon Non-TE obs (NTE) | 1321454397 | 1527967 | 3171 | 188 | 35 | 6 | 2 | 2 |
| Non-Exon Non-TE exp (NTE) | 1321451907 | 1532655 | 1206 | 0.86 | 6E−04 | 4E−07 | 3E−10 | 2E−13 |
| Exon obs (EX) | 119708384 | 97488 | 245 | 23 | 0 | 0 | 1 | 0 |
| Exon exp (EX) | 119708145 | 97939 | 57 | 0.03 | 2E−05 | 7E−09 | 3E−12 | 1E−15 |
| Total obs | 2786134823 | 3275135 | 10450 | 973 | 165 | 32 | 12 | 5 |
| Total exp | 2786124411 | 3294490 | 2692 | 2.04 | 2E−03 | 1E−06 | 8E−10 | 5E−13 |
| **(B)–Mappable 100** | | | | | | | | |
| Non-Exon TE obs (TE) | 1223239922 | 1517676 | 3927 | 266 | 25 | 11 | 5 | 1 |
| Non-Exon TE exp (TE) | 1223236637 | 1523873 | 1322 | 1.07 | 9E−04 | 7E−07 | 5E−10 | 4E−13 |
| Non-Exon Non-TE obs (NTE) | 1276165087 | 1499761 | 2698 | 97 | 16 | 2 | 0 | 1 |
| Non-Exon Non-TE exp (NTE) | 1276163336 | 1503124 | 1201 | 0.88 | 6E−04 | 5E−07 | 3E−10 | 2E−13 |
| Exon obs (EX) | 112360615 | 93084 | 185 | 16 | 0 | 0 | 0 | 0 |
| Exon exp (EX) | 112360453 | 93392 | 55 | 0.03 | 2E−05 | 7E−09 | 3E−12 | 1E−15 |
| Total obs | 2611765624 | 3110521 | 6810 | 379 | 41 | 13 | 5 | 2 |
| Total exp | 2611760426 | 3120389 | 2578 | 2 | 2E−03 | 1E−06 | 8E−10 | 6E−13 |
| **(C)–Mappable 20** | | | | | | | | |
| Non-Exon TE obs (TE) | 388613299 | 480820 | 741 | 9 | 0 | 0 | 0 | 0 |
| Non-Exon TE exp (TE) | 388612958 | 481494 | 417 | 0.34 | 3E−04 | 2E−07 | 2E−10 | 1E−13 |
| Non-Exon Non-TE obs (NTE) | 892370709 | 1061716 | 1621 | 31 | 4 | 1 | 0 | 1 |
| Non-Exon Non-TE exp (NTE) | 892369874 | 1063340 | 868 | 0.65 | 5E−04 | 3E−07 | 2E−10 | 2E−13 |
| Exon obs (EX) | 74735962 | 61034 | 103 | 6.00 | 0 | 0 | 0 | 0 |
| Exon exp (EX) | 74735883 | 61187 | 36 | 0.02 | 9E−06 | 4E−09 | 2E−12 | 7E−16 |
| Total obs | 1355719970 | 1603570 | 2465 | 46 | 4 | 1 | 0 | 1 |
| Total exp | 1355718714 | 1606021 | 1321 | 1 | 8E−04 | 6E−07 | 4E−10 | 3E−13 |

SNVs (Table 1A, Fig. 1)—the difference is highly significant using the Chi-square goodness of fit test ($p < 0.0001$) for both the whole genome (Total) and when separating the genome into non-coding transposable elements (TE), non-coding non-transposable elements and (NTE) and exons (EX) (Table 1A). We refer to sites with 3 or more SNVs as excess sites. In total we observed 1,187 excess sites (Table 1A) with the density of excess sites in TE being 3.9 and 3.4 fold greater than in NTE and EX respectively. The probability of this level of SNV recurrence by chance alone is so low (Chi-squared goodness of fit test, $p > 0.0001$) that these excess sites must either be (i) drivers, (ii) the result of mutation rate heterogeneity across the genome or, (iii) the consequence of next generation sequencing (NGS) pipeline errors.

It seems unlikely that the majority of the excess sites are due to drivers since the density of excess sites is higher in the TE and NTE parts of the genome than in EX (Table 1A). Furthermore, to date only one intergenic driver of cancer—an activating $C > T$ mutation

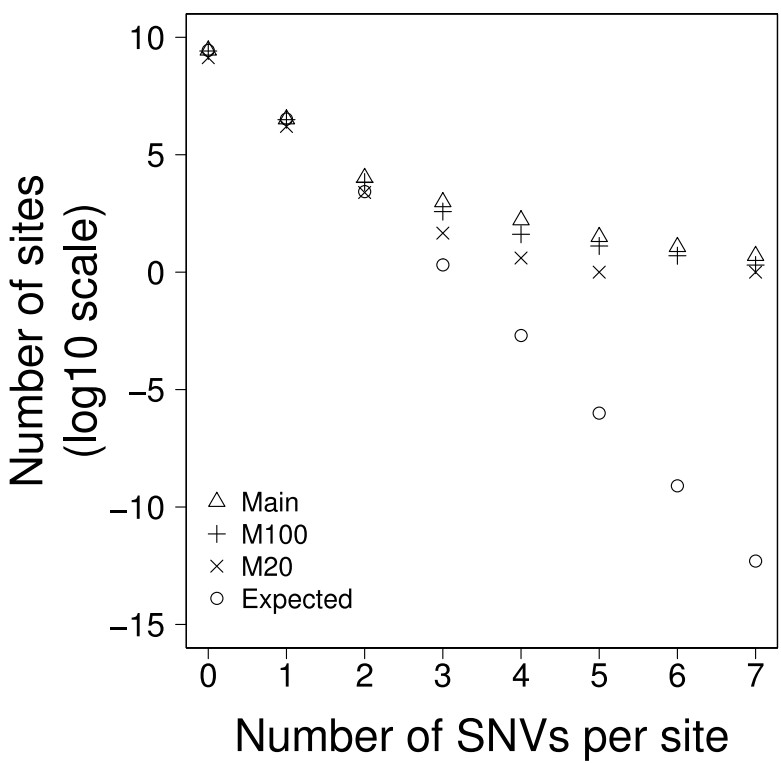

**Figure 1** The number of sites with 0–7 SNVs per site for: Main = all data, M100 = sites that are uniquely mappable at 100 base-pairs, M20 = sites that are uniquely mappable at base-pairs and, Expected is the expected number of SNVs per site drawn from a Poisson distribution using all data.

in the *TERT* promoter (*Huang et al., 2013*) at chr5:1,295,228—has been confirmed, and although this is included in the excess sites with 7 SNVs, the remaining 1,186 excess sites are unlikely to be under such selection. It therefore seems likely that the excess sites are either due to mutation rate variation or problems with sequencing.

## Excess sites are enriched in non-unique sequences

The human genome contains many duplicated sequences particularly within transposable elements, and these pose challenges for accurate alignment of the short ~100bp reads produced from NGS (*Zhuang et al., 2014*). If the excess sites were the result of NGS mapping errors then we might expect them to occur in regions of the genome that were hard to align. Using the mappability scores (*Derrien et al., 2012*) we excluded all bases that were not uniquely mappable at 100bp; this should give an overall indication of how easy it is to map reads to the region. This only reduced the interrogable genome by 6%, but the number of excess sites was reduced by 64% (Table 1B), demonstrating that a large proportion of the excess sites were in duplicated sequences and therefore likely originate from mapping errors. However, even with this large reduction in excess sites we still observed many excess sites far greater than chance expectation (Chi-squared goodness of fit test, $p < 0.0001$) (Table 1B and Fig. 1).

The SNVs in this data were all called from >100bp reads. If the excess sites were errors of read mapping, they should not be affected by the uniqueness of shorter sequences

(i.e., there is no reason why 100bp sequences that map uniquely to the genome should be mis-mapped if it contains a non-unique 20bp sequence), however if the SNVs were the product of a biological process that was more prevalent in non-unique or repetitive sequences, then we might expect to see a reduction of excess sites when we exclude all bases that do not map uniquely at 20bp. When we excluded all bases that were not unique at 20bp we found that the interrogable genome was reduced by 52% and the excess sites were reduced by 96% (Table 1C and Fig. 1). It is worth noting that, due to their proliferative nature throughout the genome, this reduction disproportionately affects TEs where the interrogable genome is reduced by 71% and the excess sites by >99%. This would suggest that the excess sites existing in sequences that were unique at 100bp but not unique at 20bp may represent some biological process and not error. Furthermore, the *TERT* promoter, whose recurrence is the result of positive selection, and is therefore the only excess site that that we can confidently say is not a product of error, remains in this most conservative of these analyses. Despite this large reduction in excess sites, significant heterogeneity still remains; the probability of observing the 52 excess sites in the part of the genome uniquely mappable at 20 bases is still extremely low (Chi-squared goodness of fit test, $p < 0.0001$).

One other potential problem with mapping reads to non-unique sequences occurs when a segmental duplication has been collapsed in the assembly of the reference genome; i.e., reads from two different locations are mapped to the same locus in the reference. Differences between the duplications will appear as SNVs. If this was the case we would expect to see an increase in excess site read coverage of ∼2-fold or greater. To investigate whether this could be a problem in our data we compared the read coverage for excess sites and non-excess sites, which nevertheless had an SNV, in the one set of cancer genomes for which we had this information—the liver cancers sequenced by the RIKEN group. However, we found that the median read coverage for the excess sites ($n = 15$) was actually lower than for non-excess sites ($n = 224, 602$) (28 and 33 reads respectively; Mann–Whitney $U$ test, $p = 0.043$).

## Privacy of mutations

To further investigate the origin of excess sites we exploited the fact that some types of cancer were sequenced by different laboratories using different technologies and NGS pipelines. If the SNVs at excess sites found in a particular cancer are due to hypermutable sites then we would expect them to be randomly distributed across research groups (i.e. all research groups should identify the same hypermutable sites). If however the SNVs at excess sites are due to error then we might expect them to be heterogeneously distributed across research groups (i.e. the calling of recurrent false positive SNVs should be systematic of individual research group NGS pipelines). The liver cancers, which were all virus associated hepatocellular carcinomas, were sequenced by two different groups; 66 from the RIKEN group using the Illumina Genome Analyser (https://dcc.icgc.org/projects/LIRI-JP) and 22 from the National Cancer Centre in Japan using the IIlumina HiSeq platform (https://dcc.icgc.org/projects/LINC-JP). We found that the excess SNVs were heterogeneously distributed amongst research groups (Fisher's exact test, $P = 4 \times 10^{-6}$) suggesting that the 30 excess sites from liver cancers were predominantly errors (Table S1).
**Table 2** **The fit of 4 models to the observed distribution of recurrent SNVs in the three different genomic fractions (A) TE, (B) NTE and (C) EX.** The median shape parameters are given for models 1b and 2b and the median eta are given for models 2b.

| Model | N | Log-likelihood | Shape | $\varepsilon$ |
|---|---|---|---|---|
| Non-Exon TE (TE) | | | | |
| 1a | 2 | −269283.00 | 0.13 | |
| 1b | 64 | −2935.80 | 0.12 | |
| 2a | 3 | −266889.00 | 0.00021 | 0.956 |
| *2b* | *96* | *−1302.31* | *0.00016* | *0.959* |
| Non-Exon Non-TE (NTE) | | | | |
| 1a | 2 | −227728.00 | 0.31 | |
| 1b | 64 | −1206.53 | 0.37 | |
| 2a | 3 | −227026.00 | 0.00039 | 0.963 |
| *2b* | *96* | *−565.92* | *0.00026* | *0.972* |
| Exon (EX) | | | | |
| 1a | 2 | −13877.9 | 0.18 | |
| 1b | 64 | −270.47 | 0.22 | |
| 2a | 3 | −13843.30 | 0.00019 | 0.966 |
| *2b* | *96* | *−240.68* | *0.00035* | *0.958* |

**Notes.**
 $N$, number of parameters.
 Italics indicate the best fit as determined by a likelihood ratio test.

## Parameter estimation

To gauge how much variation there is in the density of SNVs across the genome we fit two models to the data using maximum likelihood. In model 1 we allowed the density of SNVs to vary between sites according to a gamma distribution, estimating the shape parameter, and hence the amount of variation there was between sites. We fitted two versions of this model. In the first version, 1a, we constrained the model such that the mean SNV density, shape parameter, and hence the level of variation, was the same for all triplets. In the second version, 1b, we allowed the mean SNV density and shape parameter to vary between triplets. The second of these models fits the data significantly better than the first according to a likelihood ratio test suggesting that the level of variation differs between triplets (Table 2). However, a goodness of fit test, comparing the number of sites predicted to have 1, 2, 3…etc SNVs per site to the observed data, suggests the model fits the data poorly. We therefore fit a second pair of models in which we allowed the rate of SNVs to be due to two processes. The first process is constant across sites whereas the second process is variable and drawn from a gamma distribution. There are two parameters in the model, the proportion of SNVs at a site produced by the first process and the level of variation in the second process. This model might represent a situation where the rate of mutation is constant across sites but the rate of sequencing error is variable. As with the first model we fit two versions of this model; in Model 2a we constrained the model such that the parameters of the two processes were the same for all triplets. In Model 2b they were allowed to vary between triplets. Both models 2a and 2b fit the data significantly
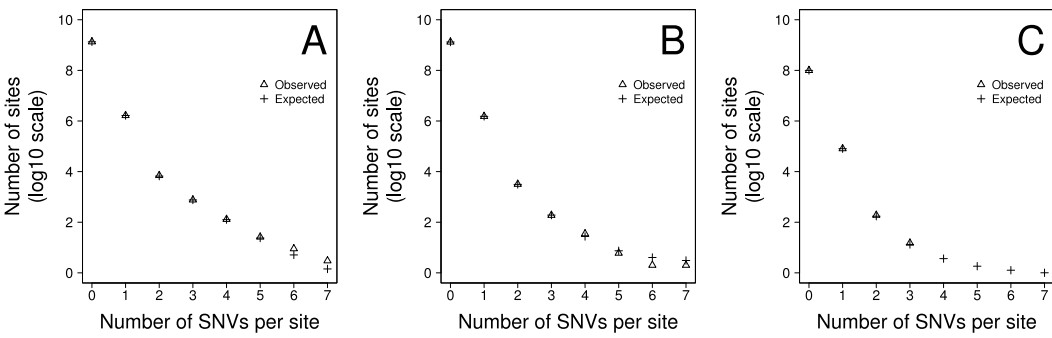

**Figure 2** The fit of the observed recurrent SNV distribution to expected distribution under the favoured model, 2b, for (A) TE, (B) NTE and (C) EX genomic fractions

better than models 1a and 1b, and of this second pair of models, model 2b, which allows the parameters to vary between triplets fits the data significantly better than model 2a, in which the parameters are shared across triplets (Table 2). The best fitting model is therefore one in which we have two processes contributing to the production of SNVs, one that is constant across sites, although it differs between triplets, and one which is variable across sites. Although, we can formally reject this model using a goodness-of-fit test (Chi-square $p < 0.0001$), because we have so much data, it is clear that the model fits the data fairly well (Fig. 2). Under this model we estimate that approximately 4.1%, 2.8% and 4.2% of SNVs are due to the process that varies across sites in the TE and NTE, and EX sequences respectively. However, the variation in the density between sites due to the variable process is extremely large. The median shape parameters are 0.00016, 0.00026 and 0.00035 for the TE and NTE, and EX sequences respectively. Under a gamma distribution with a shape parameter of 0.0004 we would expect more than 99% of sites to have no SNVs generated by this variable process, but some sites to have a density of SNVs that is 30,000-fold above the average rate.

## DISCUSSION

Through our analysis of ~3 million SNVs from whole cancer genomes we have shown that there are many sites at which there is a significant excess of SNVs. The majority of these are unlikely to be drivers because the density of sites with an excess of SNVs is greater in the non-coding part of the genome than in the exons. It therefore seems likely that the majority of the excess sites are either due to hypermutation or problems with sequencing or the processing of the sequences. Several lines of evidence point to sequencing problems being the chief culprit. First, many of the excess sites disappear when regions of the genome with low mappability are removed. Second, SNVs at a particular excess site tend to be found within the sequences from a particular laboratory; for example, site 85,091,895 on chromosome 5 has 5 SNVs in liver cancers, but all of these are found in the sequences from RIKEN not the sequences from the NCC. It is possbile that this could be caused by biological differences between the cohorts, either environmentally induced or endogenous genetic variation, such as that seen between European and African populations and the
differing frequency of 5′-TCC-3′ > 5′-TTC-3′ mutation (*Harris, 2015*). However the level of site and cohort specific, but cryptic, variation required would be huge and we have very little evidence to support such a hypothesis. Third, the level of variation in the density of SNVs is much greater than has been observed or suggested for variation in the mutation rate (*Hodgkinson & Eyre-Walker, 2011*; *Kong et al., 2012*; *Michaelson et al., 2012*) though see a recent analysis of de novo germ-line mutations which suggests there could be extreme mutational heterogeneity (*Smith et al., 2016*); some sites are estimated to have rates of SNV production that are tens of thousands of times faster than the genomic average.

Only one line of evidence suggests that there might also be substantial variation in the mutation rate as well as variation in the error rate. When we eliminate sites that are not uniquely mappable at 20bp we find a great reduction in the number of excess sites relative to the case when we remove sites that are not uniquely mappable at 100bp, and yet the read length is greater than 100bp in the data that we have used. This might suggest that there are some repetitive sequences that are prone to a process of hyper-mutation. However, it might also be that mappability at 100bp is not a good guide to mappability during sequence processing. First, some level of mismatch must be allowed during the mapping of reads to the reference because there are single nucleotide variants segregating in the population and there are somatic mutations in cancer genomes. Second, the mappability score is assigned to the first nucleotide of the $k$-mer that can be mapped. Third, although the read length was greater than 100bp, some shorter reads may have been used. Next generation sequencing involves a number of biological processes, such as the polymerase chain reactions in the pre-sequencing creation of libraries and the polymerization of nucleotides during sequencing by synthesis, any one of which can result in technology-specific sequencing artefacts (*Quail et al., 2008*; *Nazarian et al., 2010*). In addition to the considerable post-sequencing processing, such as filtering and mapping, which can also generate errors (*Harismendy & Frazer, 2009*; *Minoche, Dohm & Himmelbauer, 2011*). Unfortunately it is not possible to say which of these factors is most important.

We have fit two models to the data in which the density of SNVs varies across sites. In the first we imagine that the variation is due to a single variable process and in the second we imagine it is due to two processes, one of which is constant across sites and one which is variable. We find that this second model fits the data much better than the first model, although it can be formally rejected by a goodness-of-fit test. In this second model we estimate the proportion of SNVs that are due to the two processes and the level of variation. We estimate that approximately 2.8–4.2% of SNVs are due to the second process and that this second process is highly variable between sites, such that a few sites have a density of SNVs that is ten of thousands higher than the average density. It is possible that the first process is mutation and the second is sequencing error, but we cannot rule out the possibility that the second process includes variation in the mutation rate as well. Studies of germ-line (*Hodgkinson & Eyre-Walker, 2011*; *Michaelson et al., 2012*) and somatic (*Hodgkinson, Chen & Eyre-Walker, 2012*; *Woo & Li, 2012*; *Lawrence et al., 2013*; *Liu, De & Michor, 2013*; *Polak et al., 2015*) mutations have indicated that the mutation rate varies between sites on a number of different scales. However, indications are that the variation is probably fairly modest (*Hodgkinson, Chen & Eyre-Walker, 2012*; *Michaelson et al., 2012*).

A model including two processes fits the data well (Fig. 2). However, we can reject this model in a goodness-of-fit test, because we have a huge amount of data. Possible reasons for the less than perfect fit include large scale variation in the mutation rate (*Hodgkinson & Eyre-Walker, 2011*; *Schuster-Bockler & Lehner, 2012*; *Makova & Hardison, 2015*) and multi-nucleotide-mutations (MNMs) (*Rosenfeld, Malhotra & Lencz, 2010*; *Schrider, Hourmozdi & Hahn, 2011*; *Harris & Nielsen, 2014*); the latter represent ∼2% of all human single nucleotide polymorphisms (SNPs).

In conclusion it seems likely that many sites in somatic tissues that have experienced recurrent SNVs are due to sequencing errors or artefacts of post-sequencing processing and there seems to be little evidence of cryptic variation in the somatic mutation rate. However, this does not necessarily mean that such variation does not exist—it would be extremely difficult to detect it given the high level of site-specific sequencing error. As sequencing technology and processing pipelines improve in accuracy, we would expect similar future analyses to be able to confidently estimate the true underlying variation in the somatic mutation rate. Accompanied by the flow of data from projects such as the 100k genomes project, it should soon be possible to achieve per triplet mutation rate variation map for individual cancer types and not just pooled across multiple cancers.

### Funding

This work was financially supported by MRC grant G1100074. The funders had no role in study design, data collection and analysis, decision to publish, or preparation of the manuscript.

### Grant Disclosures

The following grant information was disclosed by the authors:
MRC: G1100074.

### Competing Interests

The authors declare there are no competing interests.

### Author Contributions

- Thomas C.A. Smith conceived and designed the experiments, performed the experiments, analyzed the data, contributed reagents/materials/analysis tools, wrote the paper, prepared figures and/or tables.
- Antony M. Carr conceived and designed the experiments, reviewed drafts of the paper.
- Adam C. Eyre-Walker conceived and designed the experiments, performed the experiments, analyzed the data, contributed reagents/materials/analysis tools, reviewed drafts of the paper.

### Data Availability

Smith, Thomas (2016): Mutation rate variation or error supplementary code and data. figshare. https://dx.doi.org/10.6084/m9.figshare.3398899.v1.

## Supplemental Information

Supplemental information for this article can be found online at http://dx.doi.org/10.7717/peerj.2391#supplemental-information.

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
