# Peer review of "Are sites with multiple single nucleotide variants in cancer genomes a consequence of drivers, hypermutable sites or sequencing errors?"

_PeerJ, doi:10.7717/peerj.2391_

## Round 0.1 · original submission · Minor Revisions

· Academic Editor

Minor Revisions

Both reviewers consider the study well designed and very interesting and I agree with them. However, reviewer 2 suggests two points that deserve further discussion: the influence of the presence of similar variants, although not identical, in the genome, and, more importantly, the possibility that genetic and environmental differences may have a role in the observed heterogeneity. Please discuss these points in the revised version.

Reviewer 1 ·

Basic reporting

No comments

Experimental design

No comments

Validity of the findings

Over the past eight years, the profusion of whole-genome and whole-exome sequencing studies has led to huge datasets of somatic mutations found in cancer samples. This is making it possible to see in unprecedented detail not only the driving events of cancer, but also the common artifacts of our sequencing technology. Distinguishing between these two classes of observations has proven to be a greater challenge than anticipated.

In particular, there has been a reliance in the field on observing recurrent mutations at the same genomic position in multiple samples. This is taken as prima facie evidence of positive selection in tumorigenesis, since the chances of observing exactly the same mutation in multiple samples is vanishingly small.

Studies such as Chang et al. PMID 26619011 and many others present lists of recurrently mutated positions, seemingly without appreciation for the possibility that some may actually be recurrent artifacts.

The current study points out that there are many more recurrently mutated sites in the genome than would be possible in a model without recurrent non-driver mutations. The authors show this in a straightforward, easily understood way. Furthermore, they point out that different laboratories in the world, analyzing the same kind of cancer, find many *different* recurrently mutated positions in the genome, consistent with a model of technology-specific artifacts.

This is an elegant fundamental advance in the analysis of somatic mutations in cancer, and a crucial warning for the field.

Additional comments

No comments

Reviewer 2 ·

Basic reporting

No comments

Experimental design

The study is well designed.

It is, however, unclear how "uniquely mappable" is defined. If it means that the 20-mer around the mutation only occur once in the genome, it would be more interesting to allow some wiggle room, i.e., the 20-mer is uniquely mappable if there is no other 20-mer in genome that matches with say 90%. See e.g. Bailey et al (2004) (pmid 11381028). This is particularly interesting if we assume that called variant is caused at the mismatch between two similar regions.

Validity of the findings

In the section "Privacy of mutations" the authors write the significant heterogeneity suggests the excess sites were predominantly error. While it is a possible explanation, it seems a bit overstated conclusion as it very well might be a biological explanation such as difference between Japanese and American patients - either genetic differences or environmental differences such as diet.

Additional comments

The authors state repeatedly that the model is rejected by a goodness-of-fit test, which means the model is not perfect. The paper would benefit from having some discussion on what the natural next step would be to improve the model.

Minor comment:
Bottom of page 11 it says p>0.0001; should probably be p<0.0001

---

## Round 0.2 · accepted · Accept

· Academic Editor

Accept

I think this is an important contribution that will stimulate methodological improvements in the computational treatment of NGS data and help to clarify mutational processes.